# Tadalafil Treatment Ameliorates Hypoxia and Alters Placental Expression of Proteins Downstream of mTOR Signaling in Fetal Growth Restriction

**DOI:** 10.3390/medicina56120722

**Published:** 2020-12-21

**Authors:** Kyoka Tsuchiya, Kayo Tanaka, Hiroaki Tanaka, Shintaro Maki, Naosuke Enomoto, Sho Takakura, Masafumi Nii, Kuniaki Toriyabe, Shinji Katsuragi, Tomoaki Ikeda

**Affiliations:** Department of Obstetrics and Gynecology, Mie University School of Medicine, Edobashi, Tsu 8858507, Mie, Japan; ktsuchiya1020@gmail.com (K.T.); tagami.t.ky@gmail.com (K.T.); mabochikin519@yahoo.co.jp (S.M.); nao-e@clin.medic.mie-u.ac.jp (N.E.); s-takakura@clin.medic.mie-u.ac.jp (S.T.); m-nii1984@clin.medic.mie-u.ac.jp (M.N.); to.kuniaki@gmail.com (K.T.); skatsura@clin.medic.mie-u.ac.jp (S.K.); t-ikeda@clin.medic.mie-u.ac.jp (T.I.)

**Keywords:** fetal growth restriction, placenta, mTOR

## Abstract

*Background and Objectives:* Fetal growth restriction (FGR) is associated with fetal mortality and is a risk factor for cerebral palsy and future lifestyle-related diseases. Despite extensive research, no effective treatment strategy is available for FGR. Mammalian target of rapamycin (mTOR) signaling is important for the growth of fetal organs and its dysregulation is associated with miscarriage. Here, we focused on mTOR signaling and investigated how the activities of phospho-ribosomal protein S6 (rps6) and phospho-eukaryotic translation initiation factor 4E (eIF-4E), which act downstream of mTOR signaling in the human placenta, change following treatment of FGR with tadalafil and aimed to elucidate the underlying mechanism of action. Placental hypoxia was investigated by immunostaining for hypoxia-inducible factor (HIF)-2α. *Materials and Methods:* Phosphor-rps6 and phosphor-eIF4E expression were examined by Western blotting and enzyme-linked immunosorbent assay, respectively. *Results:* HIF-2α expression significantly increased in FGR placenta compared with that in the control placenta but decreased to control levels after tadalafil treatment. Levels of phospho-rps6 and phospho-eIF-4E were significantly higher in FGR placenta than in control placenta but decreased to control levels after tadalafil treatment. *Conclusions:* Tadalafil restored the levels of HIF-2α, phospho-rps6, and eIF-4E in FGR placenta to those observed in control placenta, suggesting that it could be a promising treatment strategy for FGR.

## 1. Introduction

Fetal growth restriction (FGR) has been associated with fetal mortality [1] and serves as a risk factor for cerebral palsy, mental retardation, pervasive developmental disorder, and future lifestyle-related diseases [2,3,4]. Various treatment/prevention methods have been tested, but none proved effective [1]. FGR is divided into early-onset type, related to impairment in placental perfusion, and late-onset type, associated with impairment of placental diffusion [5]. We have conducted clinical and basic research on use of tadalafil for the treatment of early-onset FGR [6,7,8,9,10,11,12]. Molecule weight is 389.404 g/mol. Tadalafil cross the placenta from mother to fetus. Previous studies have predominantly employed sildenafil citrate as a phosphodiesterase type (PDE) 5 inhibitor. In contrast, the present study used tadalafil, which has a longer half-life (14–15 h vs. 2–4 h) and would therefore presumably result in improved stability and thus effectiveness. Another benefit of tadalafil’s longer half-life is that one dose per day would suffice, whereas sildenafil citrate would need to be administered at least twice per day [13]. Although PDE5 enzymes are widely distributed in blood vessels, tadarafil is particularly selective for PDE5 enzymes found in the reproductive organs. Although these studies could confirm the effect of tadalafil on FGR, the underlying mechanism of action could not be fully elucidated.

Various regulatory factors such as oxygen and nutrition are involved in fetal development, and mammalian target of rapamycin (mTOR) signaling is one of the major regulatory factors [13,14,15,16,17]. mTOR signaling, which targets serine/threonine kinase, is a central regulator of cell metabolism, growth, and survival in response to hormones, growth factors, nutrients, energy, and stress signals [14,15,16,17,18]. Therefore, mTOR signaling is important for the growth of fetal organs, and its dysregulation may lead to miscarriage or recurrent miscarriage.

In this study, we investigated how tadalafil treatment affects mTOR signaling in FGR placenta and the underlying mechanism of action. For this, we examined the tadalafil-induced changes in the activities of phospho-ribosomal protein S6 (rps6) and phospho-eukaryotic translation initiation factor 4E (eIF-4E), which act downstream of mTOR signaling in the human placenta.

## 2. Materials and Methods

### 2.1. Collection of Placental Samples

Study population recruitment and collection of placental tissue were performed with informed consent, which was approved by the Clinical Research Ethics Review Committee of Mie University Hospital (approval no. 3054, the day of approval: 8 November 2016). Written informed consent was obtained from a Japanese pregnant woman. The placental sample in FGR and FGR + tadalafil cases used case in clinical trial [19]. Control case selected randomly the normal case during clinical trial. The collected placenta was divided and either cryopreserved at −80 °C or fixed.

### 2.2. Histological Analysis and Immunohistochemistry (IHC) of Placenta

Placentas were fixed in 4% paraformaldehyde (Nacalai Tesque, Kyoto, Japan) in 0.2 M phosphate-buffered saline (PBS) (pH 7.4) and then embedded in paraffin (Merck Ltd., Darmstadt, Germany) using standard procedures. Placental sections were incubated at room temperature overnight with an anti-hypoxia-inducible factor (HIF)-2α antibody (Abcam, Cambridge, UK). Sections were then probed with a goat anti-rabbit IgG for 2 h and later incubated with a peroxidase–anti-peroxidase complex for 2 h. Sections were treated with Peroxidase Stain DAB Kit (Nacalai Tesque, Kyoto, Japan) and scanned using an Olympus BX60 microscope (Olympus, Tokyo, Japan). IHC results were qualitatively estimated by 4 blinded individuals. Chromogenic signal intensity was assigned a relative score of 0 (no signal), 1 (very weak signal), 2 (weak signal), 3 (moderate), 4 (strong), or 5 (very strong). The scores of 4 fields of view were averaged for each slide.

### 2.3. Protein Extraction

Cryopreserved placental samples were homogenized by T 10 basic ULTRA-TURRAX (IKA Japan K.K., Japan) in a lysis buffer (50 mM Tris-HCl, 150 mM sodium chloride (NaCl), 0.1% sodium dodecyl sulfate (SDS), 1% Nonidet P40, 1% sodium deoxycholate, and protease phosphatase inhibitor cocktail) and lysed for 30 min on ice. Samples were centrifuged at 15,000× *g* for 30 min at 4 °C, and the obtained supernatants were used as placental proteins for analysis. The total protein content was measured as per the bicinchoninic acid method using a Protein Assay BCA Kit (Nacalai Tesque, Kyoto, Japan).

### 2.4. Western Blot Analysis

Placental proteins separated on 10–20% Bis-Tris gels (ATTO, Tokyo, Japan) were transferred onto polyvinylidene fluoride membranes (ATTO). The membranes were incubated with the following primary antibodies: rabbit anti-ribosomal protein S6, anti-phospho-ribosomal protein S6 (Ser235/236) (Cell Signaling Technology, Danvers, MA, USA), and mouse anti-β-actin (Santa Cruz Biotechnology, USA). Horseradish peroxidase-conjugated secondary anti-rabbit antibodies (SeraCare Life Sciences, Milford, MA, USA) were used as secondary antibodies. Proteins were visualized using chemiluminescence detection. Densitometry analysis was performed using NIH’s ImageJ software (Wayne Rasband, Bethesda, MD, USA).

### 2.5. Enzyme-Linked Immunosorbent Assay

Phosphorylated elF-4E levels in the placenta were determined using a Human Phospho-eIF4E(S209) ELISA kit, according to the manufacturer’s instructions (RayBiotech Life, Peachtree Corners, GA, USA). Each sample was applied at a total protein concentration of 300 μg/well.

### 2.6. Statistical Analysis

Fisher’s exact test and Mann–Whitney *U* test were used to derive *p*-values for the comparison of clinical characteristics. Analysis of variance (ANOVA, Coppell, TX, USA) with Tukey–Kramer test was used for comparisons of experimental data. All analyses were performed using GraphPad Prism 7 (GraphPad, San Diego, CA, USA). A value of *p* < 0.05 was considered statistically significant.

## 3. Results

### 3.1. Clinical Characteristics

Table 1 shows the clinical data of the samples used in this study. The placentas used in this study were categorized into three groups. The FGR + tadalafil group (*n* = 12) included FGR placentas that were treated with tadalafil during the pregnancy period. The FGR group (*n* = 10) received no tadalafil treatment. The control group (*n* = 14), on the other hand, was without FGR and was managed for pregnancy and delivery at the same period in the institution.

Gestational age at birth and birth weight were significantly lower in the FGR and FGR + tadalafil groups than in the control group, and did not significantly differ between the FGR and FGR + tadalafil groups. Placental weight was significantly lower in FGR and FGR + tadalafil groups than in the control group and significantly differed between FGR and FGR + tadalafil groups. (Table 1).

### 3.2. Effect of Tadalafil Treatment on mTOR Signaling in FGR Placenta

The hypoxic conditions in the placentas were investigated through the evaluation of the expression of HIF-2α (Figure 1). The signal intensity of HIF-2α protein was significantly higher in the FGR group than in the control group (control 2.3 ± 0.18 and FGR 4.1 ± 0.15, *p* < 0.01). In contrast, HIF-2α protein expression significantly decreased in the FGR + tadalafil group as compared with that in the FGR group (FGR 4.1 ± 0.15 and FGR + tadalafil 2.3 ± 0.17, *p* < 0.01), and no significant difference was observed between the FGR and control groups (*p* > 0.99).

### 3.3. Effect of Tadalafil Treatment on rpS6 and eIF-4E in FGR Placentas

We evaluated the effects on mTOR signaling in the placenta samples from the control, FGR, and FGR + tadalafil groups. Placental mTOR signaling activity was measured by quantifying the ratio of protein levels of rpS6 and eIF-4E, which act downstream of mTORC1 (Figure 2).

No significant difference was observed in total rpS6 protein expression among the three groups (control: 1.167 ± 0.08772, FGR: 0.9043 ± 0.1084, FGR + tadalafil: 1.132 ± 0.0789; one-way ANOVA *p* = 0.1359; Figure 3B). On the other hand, the expression of phosphorylated rpS6 (Ser235/236) was significantly higher in the placenta samples from the FGR group than that in the placenta samples from the control group (control: 0.659 ± 0.06485, FGR: 1.32 ± 0.09076, FGR + tadalafil: 0.890 ± 0.06799; one-way ANOVA *p* < 0.01, post hoc Tukey *p* < 0.01 for control versus FGR, *p* = 0.06 for control versus FGR + tadalafil, and *p* < 0.01 for FGR versus FGR + tadalafil; Figure 3C).

The expression of phosphorylated elF-4E (Ser209) was significantly higher in the placenta of the FGR group than in the placenta of the control group and decreased to the level of the control group in the FGR + tadalafil group (control: 1.05 ± 0.0645, FGR: 1.63 ± 0.115, FGR + tadalafil: 1.11 ± 0.114; one-way ANOVA *p* < 0.01, post hoc Tukey *p* < 0.01 for control versus FGR, *p* = 0.89 for control versus FGR + tadalafil, *p* < 0.01 for FGR versus FGR + tadalafil; Figure 4).

## 4. Discussion

In this study, we investigated mTOR signaling in the placenta to elucidate the mechanism of action of tadalafil in FGR. We established two findings. First, FGR placenta showed increased expression of the hypoxic marker HIF-2α, and tadalafil treatment reduced the expression of HIF-2α to the level observed in the control placenta. Second, the protein expression of phospho-rps6 and phospho-eIF-4E, which act downstream of mTOR, was upregulated in FGR, and tadalafil could ameliorate this upregulation to the level observed in the control group.

In previous studies, HIF-2α expression was found to be significantly reduced in the placenta of an FGR mouse model as compared to that in the control mice, and tadalafil administration could restore these reduced levels of HIF-2α [12]. This observation was consistent with that reported in the present study, wherein human placenta increased the expression of hypoxic markers after FGR and tadalafil treatment was able to alleviate this upregulated expression of HIF-2α.

mTOR signaling is involved in the regulation of various processes and performs important functions in protein translation, autophagy, cytoskeleton regulation, and gene transcription. It is also involved in fundamental biological phenomena such as cell death, growth, and division [20]. mTOR signaling is activated under eutrophic conditions and regulates various cellular functions such as cell growth, metabolism, and protein synthesis [21]. On the other hand, it is inactivated under starvation conditions [22]. In FGR, placental hypoplasia causes a decrease in mTOR activity in the placenta and decidua, resulting in a decrease in the placental blood flow [22,23,24]. The phosphorylation of insulin-like growth factor-binding protein 1 (IGFBP-1) is reported to be increased, followed by a decrease in placental nutrient transport and trophoblast invasion and placental growth, thus contributing to FGR [22,23,24]. A study reported an increase in mTOR activity in FGR [25]. These authors evaluated the levels and localization of System L, alanine-serine-cysteine transporter 2 (ASCT2), and mTOR in the placental syncytiotrophoblast during FGR and preeclampsia and found a significant increase in the expression of System L amino acid transport proteins 4F2 cell-surface antigen heavy chain (4F2hc) and L-type amino acid transporter (LAT1) in FGR placenta compared with that in normal placenta. It was considered that mTOR activity was increased in FGR [25]. In this study, the mTOR activity was thought to be increased owing to a significant increase in the expression of the downstream phospho-rps6 and phospho-eIF-4E proteins. A study from Japan also revealed the increased expression of proteins acting downstream of mTOR signaling. FGR standards differ between Japan and Europe and the United States [1,26,27]. The Japanese standard involves measurement of the estimated body weight by ultrasonography. FGR is confirmed if the estimated body weight is less than −1.5 standard deviations (SD) [26]. In Europe and the United States, FGR is confirmed when the estimated body weight and abdominal circumference are below 10 percentile and the umbilical artery blood flow waveform is abnormal by ultrasonography [1,27]. FGR is diagnosed earlier by the Japanese standard than by the European and U.S. standards. In other words, the results of the present study may be associated with early FGR. In general, the progression of FGR pathology is slow, and mTOR activity may vary depending on the stage observed. The result of FGR recovering to the same level as the control group following management and treatment corresponds to the state where the increase in fetal weight was only −1.5 SD. In addition, the placental weight was significantly higher for the target cases in this study than that for those treated with tadalafil. Placental growth is reported to decrease following suppression of mTOR signaling. Our study shows the placental growth-promoting effects of tadalafil, in contrast to previous reports related to mTOR downstream pathways.

## 5. Conclusions

The mechanism of action of tadalafil in FGR is unclear. Administration of tadalafil to FGR led to the restoration of the activities of hypoxic markers HIF-2α, phospho-rps6, and eIF-4E to the levels observed in the control. Further studies are warranted to elucidate the underlying mechanism of action.

## Figures and Tables

**Figure 1 medicina-56-00722-f001:**
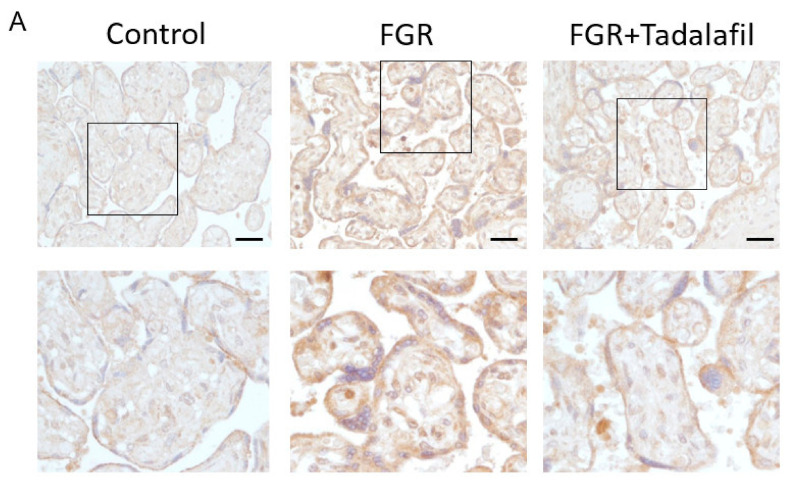
Effects of fetal growth restriction (FGR) and tadalafil on hypoxia-inducible factor (HIF)-2α protein expression in the placenta. (**A**) Representative photographs of immunohistochemistry (IHC) labeled with HIF-2α antibody in the placenta samples from the three groups (control, FGR, and FGR + tadalafil; scale bar, 50 µm). The areas in the small square are shown at higher magnification in the row below. Nuclei were counterstained with hematoxylin D. (**B**) Estimated HIF-2α IHC scoring data to define 5/95 percentile (capped bar), second and third quartiles (box), mean (plus), and outliers (dot), which are shown in a box and whisker format (*n* = 32).

**Figure 2 medicina-56-00722-f002:**
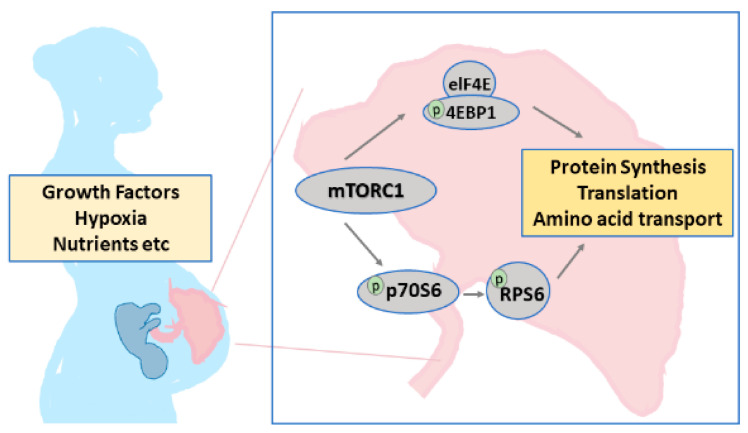
Representation of placental mammalian target of rapamycin (mTOR) signaling. Placental mTOR signaling changed downstream of mechanistic target of rapamycin complex 1 (mTORC1) in response to oxygen and nutritional status. The proteins rpS6 and eIF-4E, which act downstream of mTOR signaling, were involved in protein synthesis, translation, and amino acid transport.

**Figure 3 medicina-56-00722-f003:**
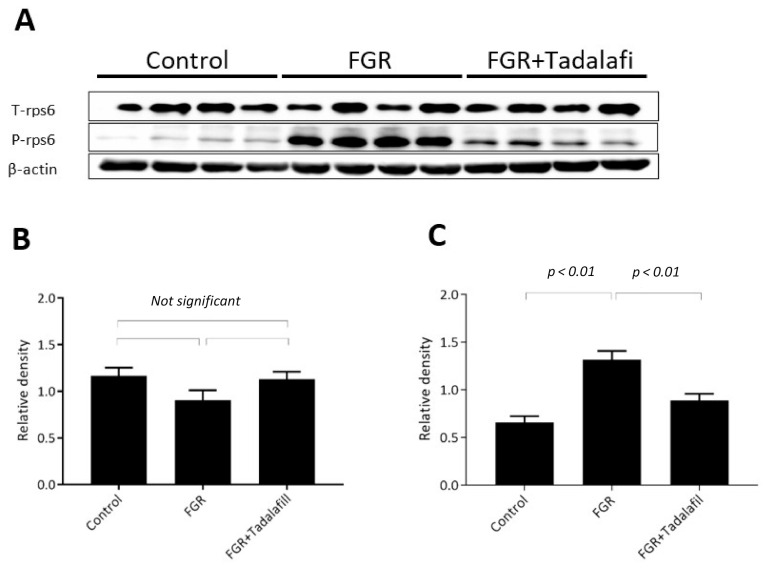
Effects of FGR and tadalafil on rps6 expression in the placenta. (**A**) Representative Western blotting images. (**B**) Relative density of total rps6 expression. (**C**) Relative density of phospho-rps6 expression. Values are means ± standard error (SE).

**Figure 4 medicina-56-00722-f004:**
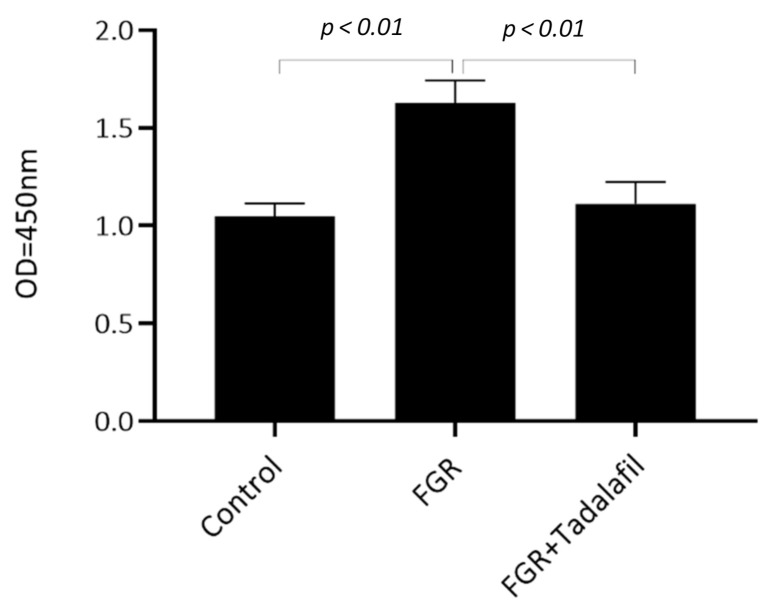
Effects of FGR and tadalafil on phosphorylated eIF4E (Ser209) expression in the placenta. Values are means ± standard error (SE).

**Table 1 medicina-56-00722-t001:** Background.

	Control (*n* = 14)	FGR (*n* = 10)	FGR + Tadalafil (*n* = 12)
Maternal age (year)	32.5 ± 1.0	35.8 ± 0.8	32.9 ± 1.2
Primiparity	7 (50%)	6 (60%)	5 (45%)
Gestational age at birth (week)	38.1 ± 0.4	35.3 ± 0.5	36.7 ± 0.6
Early onset FGR	-	10 (100%)	12 (100%)
Maternal complication	0 (0%)	0 (0%)	0 (0%)
Obstetric event without FGR	0 (0%)	0 (0%)	0 (0%)
Delivery mode			
Caesarian delivery	11 (78%)	8 (80%)	6 (45%)
Vaginal delivery	3 (22%)	2 (20%)	5 (55%)
Sex of newborn infant			
Male	6 (43%)	5 (50%)	5 (42%)
Female	8 (57%)	5 (50%)	7 (58%)
Birth weight (g)	2955 ± 68	1702 ± 80	1903 ± 96
Standard Deviation of birth weight	0.1 ± 0.5	−1.9 ± 0.3	−1.8 ± 0.3
Placental weight (g)	491 ± 21	322 ± 25	417 ± 23

FGR: Fetal growth restriction.

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
