# Peer review of "Tadalafil Treatment Ameliorates Hypoxia and Alters Placental Expression of Proteins Downstream of mTOR Signaling in Fetal Growth Restriction"

_medicina, 2020, doi:10.3390/medicina56120722_

Round 1

Reviewer 1 Report

Dear Author

The manuscript is interesting and the main finding should lead to other researches that could underline the need for maternal therapy in case of fetal grwoth restriction.

There are some comments I would like to submit:

the main finding is the effect of taladafil on HIF-2alfa, that is why it should be better specify

1-which kind of molecule is taladafil, which are its common use and which are the known side effects on humans

2- why taladafil was choosen ( vs sildenafil for example)

3- why HIF-2 alfa was considered as a optimal marker for hypoxya: are there any international consensus? Is it comparable to other similar marker? Is this measurement reproducible?

4- were there any correlation between Apgar score at birth, neonatal acidosis and placental hypoxya: in other words, did taladafil ameliorate neonatal outcome apart of level placental oxygenation?

5- could the author justify how did they decide to submit tadalafil or not? Since there was not a randomization, which were the criteria of inclusion in one group or in another?

Author Response

December 18th, 2020,

Manuscript ID: medicina-1015538

Medicina

Dear Editor

We greatly appreciate the reviewers’ constructive comments. According to the comments, we have revised our manuscript according to the comment and suggestions, as shown below. We hope that the revised version meets all of their requests.

With best regards,

Dr. Hiroaki Tanaka

Department of Obstetrics and Gynecology, Mie University School of Medicine

2-174 Edobashi, Tsu, Mie, Japan

Tel: +81-059-232-1111

Fax: +81-059-231-5202

E-mail:h_tanaka@med.miyazaki-u.ac.jp

Reviewer 1

1-which kind of molecule is taladafil, which are its common use and which are the known side effects on humans

Response; We added the sentence in Introduction section as follow, ‘Molecule weight is 389.404 g/mol. Tadalafil cross the placenta from mother to fetus.’.

2- why taladafil was choosen ( vs sildenafil for example)

Response; We added the sentence in introduction sentence as follow, ‘Previous studies have predominantly employed sildenafil citrate as a PDE5 inhibitor. In contrast, the present study used tadalafil, which has a longer half-life (14-15 hours vs 2-4 hours), and would therefore presumably result in improved stability and therefore effectiveness. Another benefit of tadalafil’s longer half-life is that one dose per day would suffice, whereas sildenafil citrate would need to be administered at least twice per day. Although PDE5 enzymes are widely distributed in blood vessels, tadarafil is particularly selective for PDE5 enzymes found in the reproductive organs .’

3- why HIF-2 alfa was considered as a optimal marker for hypoxya: are there any international consensus? Is it comparable to other similar marker? Is this measurement reproducible?

Response; We have not measure HIF-1α in placenta with the emphasis on previous report※. HIF-2α positive cells was endothelial cells, neuronal cells, and placental cells. HIF-2α was localized in multiple cells of placenta.

※Fujii, T. et al. Enhanced HIF2α expression during human trophoblast differentiation into syncytiotrophoblast suppresses transcription of placental growth factor. Sci. Rep. 2017, 7, 12455.

4- were there any correlation between Apgar score at birth, neonatal acidosis and placental hypoxya: in other words, did taladafil ameliorate neonatal outcome apart of level placental oxygenation?

Response; We have showed prolongation of pregnancy periods due to tadalafil in RCT※.

※ Maki S, Tanaka H, Tsuji M, et al. Safety Evaluation of Tadalafil Treatment for Fetuses with Early-Onset Growth Restriction (TADAFER): Results from the Phase II Trial. J Clin Med. 2019, 8, 856.

5- could the author justify how did they decide to submit tadalafil or not? Since there was not a randomization, which were the criteria of inclusion in one group or in another?

Response; We have used the FGR cases randomized in RCT※.

※ Maki S, Tanaka H, Tsuji M, et al. Safety Evaluation of Tadalafil Treatment for Fetuses with Early-Onset Growth Restriction (TADAFER): Results from the Phase II Trial. J Clin Med. 2019, 8, 856.

Reviewer 2 Report

This report investigated how tadalafil treatment affects mTOR signaling in FGR placenta and the underlying mechanism of action.
Their results provide useful information for understanding pathophysiology and establishing therapeutic strategies for FGR.

I have some comments.

Please clarify the details of tadalafil administration and the definition of FGR in this study by providing references.

For birth weight and placental weight in Table 1, please also compare SD values if possible.

Please also list in Table 1 the gestational age of FGR diagnosed or the percentage of early-onset FGR.

P3 L108-109
“and significantly differed between FGR and FGR + Tadalafil groups. Placental weights were significant lower in FGR group than in FGR + Tadalafil group (Table 1).”
This parts state the same thing, please combine them into one.

Author Response

December 18th, 2020,

Manuscript ID: medicina-1015538

Medicina

Dear Editor

We greatly appreciate the reviewers’ constructive comments. According to the comments, we have revised our manuscript according to the comment and suggestions, as shown below. We hope that the revised version meets all of their requests.

With best regards,

Dr. Hiroaki Tanaka

Department of Obstetrics and Gynecology, Mie University School of Medicine

2-174 Edobashi, Tsu, Mie, Japan

Tel: +81-059-232-1111

Fax: +81-059-231-5202

E-mail:h_tanaka@med.miyazaki-u.ac.jp

Reviewer 2

Please clarify the details of tadalafil administration and the definition of FGR in this study by providing references.

Response; We have added the sentence and reference in Methods section as follow. ‘The placental sample in FGR and FGR + tadalafil cases used case in clinical trial [19]. Control case selected randomly the normal case during clinical trial.’

For birth weight and placental weight in Table 1, please also compare SD values if possible.

Response; We added the SD values of birth weight.

Please also list in Table 1 the gestational age of FGR diagnosed or the percentage of early-onset FGR.

Response; We added the percentage of early-onset FGR in Table 1. Gestational age of FGR diagnosed can’t re-investigated for anonymized data.

P3 L108-109

“and significantly differed between FGR and FGR + Tadalafil groups. Placental weights were significant lower in FGR group than in FGR + Tadalafil group (Table 1).”

This parts state the same thing, please combine them into one.

Response; We delated the sentence pointed out.